# Association between single nucleotide polymorphisms (SNPs) of IL1, IL12, IL28 and TLR4 and symptoms of congenital cytomegalovirus infection

**Dominika Jedlińska-Pijanowska**[1], **Beata Kasztelewicz**[2], **Justyna Czech-Kowalska**[1]*, **Maciej Jaworski**[3], **Klaudia Charusta-Sienkiewicz**[1], **Anna Dobrzańska**[1]

**1** Neonatal Intensive Care Unit, The Children's Memorial Health Institute, Warsaw, Poland, **2** Department of Clinical Microbiology and Immunology, The Children's Memorial Health Institute, Warsaw, Poland, **3** Department of Biochemistry, Radioimmunology and Experimental Medicine, The Children's Memorial Health Institute, Warsaw, Poland

☯ These authors contributed equally to this work.
* j.kowalska@ipczd.pl

## Abstract

Congenital cytomegalovirus (cCMV) infection is the most common intrauterine infection. A non-specific immune response is the first line of host defense mechanism against human cytomegalovirus (HCMV). There is limited data on associations between Single Nucleotide Polymorphisms (SNPs) in genes involving innate immunity and the risk and clinical manifestation of cCMV infection. The aim of the study was to investigate association between selected SNPs in genes encoding cytokines and cytokine receptors, and predisposition to cCMV infection including symptomatic course of disease and symptoms. A panel of eight SNPs: IL1B rs16944, IL12B rs3212227, IL28B rs12979860, CCL2 rs1024611, DC-SIGN rs735240, TLR2 rs5743708, TLR4 rs4986791, TLR9 rs352140 was analyzed in 233 infants (92 cCMV-infected and 141 healthy controls). Associations between genotyped SNPs and predisposition to cCMV infection and symptoms were analyzed. The association analysis was performed using SNPStats software. No statistically significant association was found between any genotyped SNPs and predisposition to cCMV infection and symptomatic course of disease. In relation to particular symptoms, polymorphism of IL12B rs3212227 was linked to decreased risk of prematurity (OR = 0.37;95%CI,0.14–0.98;p = 0.025), while polymorphism of IL1B rs16944 was linked to reduced risk of splenomegaly (OR = 0.36;95% CI,0.14–0.98; p = 0.034) in infants with cCMV infection. An increased risk of thrombocytopenia was associated with IL28B rs12979860 polymorphism (OR = 2.55;95%CI,1.03–6.32;p = 0.042), while hepatitis was associated with SNP of TLR4rs4986791 (OR = 7.80;95% CI,1.49–40,81; p = 0.024). This is the first study to demonstrate four new associations between SNPs in selected genes (IL1B, IL12B, IL28B, TLR4) and particular symptoms in cCMV disease. Further studies on the role of SNPs in the pathogenesis of cCMV infection and incorporation of selected SNPs in the clinical practice might be considered in the future.

**Data Availability Statement:** All relevant data are within the paper and its Supporting Information file.

**Funding:** Work was supported by internal research grant of the Children's Memorial Health Institute no S158/17.

**Competing interests:** The authors have declared that no competing interests exist.

## Introduction

Congenital cytomegalovirus (cCMV) infection is the most common intrauterine infection. The HCMV is responsible for approximately 20,000–40,000 congenital infections in the United States each year [1, 2]. Nevertheless, only 10–15% of cCMV-infected newborns present clinical findings after birth, while almost 85–90% are asymptomatic [3]. Both groups are at a high risk of developing long-term sequelae. It concerns about 40–65% of symptomatic and up to 23% of asymptomatic neonates after birth [4, 5].

The clinical manifestations of cCMV disease include intrauterine growth restriction (IUGR), sensorineural hearing loss (SNHL), chorioretinitis and central nervous system (CNS) disorders (microcephaly, seizures, abnormal neuroimaging findings). Hepatobiliary and reticuloendothelial system abnormalities include hepatomegaly, splenomegaly, petechiae, thrombocytopenia, neutropenia, hepatitis and cholestasis. Abnormalities of the hepatobiliary and reticuloendothelial system seem to be transient, while SNHL and CNS disorders tend to be persisted and most devastating to the children [6].

The first line of host defense mechanisms against HCMV is a non-specific (innate) immune response. Single Nucleotide Polymorphisms (SNPs) in genes involving innate immunity has been quite extensively studied in the context of HCMV infections in organ transplant patients and stem cells recipients [7–9]. There is limited data concerning cCMV infection so far. A few previous reports linked SNP polymorphism to the risk of cCMV infection and clinical manifestation of the disease [10–13].

The aim of the study was to investigate association between eight SNPs in candidate genes encoding cytokines and cytokine receptors (IL1B rs16944, IL12B rs3212227, IL28B rs12979860, CCL2 rs1024611, DC-SIGN rs735240, TLR2 rs5743708, TLR4 rs4986791, TLR9 rs352140) and predisposition to cCMV including symptomatic course of the disease and particular symptoms.

## Material and methods

### Study population

We prospectively enrolled the Caucasian neonates with cCMV (cCMV group) and neonates without cCMV infection (healthy control group) hospitalized in NICU of The Children's Memorial Health Institute in Warsaw between March 2016 and January 2019.

Inclusion criteria for cCMV group were positive real-time PCR DNA CMV in urine $\leq 21^{st}$ day of life and parental consent. Exclusion criteria for cCMV group were: multiple congenital anomalies, severe course of bacterial septic disease, other TORCH infections (e.g. congenital toxoplasmosis), lack of parental consent. Inclusion criteria for control group were: negative real-time PCR DNA CMV in urine and parental consent. Exclusion criteria for control group were: positive real-time PCR DNA CMV in urine, multiple congenital syndromes, severe course of bacterial septic disease, other TORCH infections, lack of parental consent.

All neonates with cCMV infection underwent complete physical examination, ophthalmologic and hearing evaluation [newborn hearing screening tests (Otoacoustic Emission, OAE and Auditory Brainstem Response, ABR)]. Additionally, neuroimaging [brain ultrasounds (US) and magnetic resonance imaging (MRI)], and laboratory blood tests (bone marrow function and hepatic parameters) were performed during diagnostic process in NICU.

Patient with symptomatic cCMV infection was defined as newborn who presented at least one of the following HCMV-related signs at birth: microcephaly, abnormal hearing (SNHL was defined as air conduction thresholds > 20dBHL on the ABR with normal bone conduction thresholds and normal middle ear function), chorioretinitis, any neuroimaging findings

in MRI (e.g. ventriculomegaly, white matter abnormalities, cerebral cortex defect, peri/intra-ventricular cyst, intracerebral calcifications, myelin disorders) or at least three abnormalities of hepatobiliary and reticuloendothelial system. Following definitions for laboratory abnormalities were adopted in the study: elevated direct bilirubin >1 mg/dl for cholestasis, elevated level of aspartate transaminase (ASPAT) >84 U/l and/or elevated level of alanine aminotransferase (ALTAT) >60 U/l for hepatitis, thrombocytopenia below 100 K/μl, neutropenia below 1000 K/μl. We did not consider infants with an abnormal muscle tone, seizures, intrauterine growth restriction (IUGR) or prematurity (gestational age <37 weeks) alone as having symptomatic cCMV infection.

## Ethics statement

This study protocol was approved by the Ethics Committee of The Children's Memorial Health Institute on human subjects (28/KBE/2016 and 10/KBE/2017). We obtained written informed consent from parents of each participant before enrollment into the study. We declare compliance with ethical practices in the study.

## Detection of HCMV DNA

Total genomic DNA was extracted from 200 μL of clinical specimen (whole blood and urine) using a Nucleosp in Tissue (Macherey-Nagel, Duren, Germany) on QIAcube instrument (Qiagen, Hilden, Germany) with final elution volume of 100 μL according to manufacturer's instructions. An internal control was included during sample and negative control preparation. The presence of HCMV DNA was evaluated primarily in urine samples. Newborns with cCMV infection had HCMV DNA tested in whole blood. The detection of HCMV DNA was performed by qualitative real-time PCR as described previously [13].

## Candidate genes selection

SNPs in eight candidate genes were selected *a priori* based on previously reported associations in HCMV infections [7, 8, 10, 13–16]. PPRs such as TLR-2 (Toll-like receptor), TLR-4, TLR-9 and DC-SIGN (dendritic cell-specific ICAM-grabbing non-integrin) were chosen due to their role of HCMV recognition and triggering inflammatory cytokines [10, 11, 17–20]. Interleukins such as IL-1B, IL-12B and IL-28B were chosen for their pro-inflammatory and immunomodulatory role in the context of HCMV infection [13, 15, 21, 22]. Finally, we selected CCL 2 (C-C motif chemokine ligand 2) which was a monocyte chemotactic protein–1 (MCP-1) responsible for immune cells differentiation and their migration during inflammation in HCMV infection [23].

## Determination of SNP genotypes

SNPs genotyping was performed using genomic DNA extracted from whole blood collected for detection of HCMV. No additional sample of blood from newborn was demanded for the study. Briefly, TaqMan SNP Predesigned Genotyping Assays (Applied Biosystems, Inc., Foster City, CA, USA) were applied for IL1B rs16944, IL12B rs3212227, IL28B rs12979860, CCL2 rs1024611, DC-SIGN rs735240, TLR2 rs5743708, TLR4 rs4986791, TLR9 rs352140 polymorphisms. The allelic discrimination was performed on the 7500 Real-time PCR System (Applied Biosystems) according to the manufacturer's instructions. A blinded duplicated genotyping of 10 random study samples demonstrated 100% concordance.

## Statistical analyses

The normality of the distribution of analyzed data was assessed by Shapiro-Wilk test. Departures from Gaussian distribution were ascertained. Thus, data were presented as median and quartiles and non-parametric Mann-Whitney test was used. Counts were presented as numbers and percentages and Chi squared test was used. In the case of expected number of observations less than 5, Fisher's exact test was used. The association analysis of eight selected SNPs in following configurations was calculated: cCMV-infected/uninfected, symptomatic/asymptomatic. Among symptomatic infants following symptoms were analyzed: microcephaly, chorioretinitis, abnormal hearing, abnormal MR imaging, hepatomegaly, splenomegaly, petechiae, cholestasis, thrombocytopenia, neutropenia, hepatitis, premature and IUGR. All SNPs were analyzed for Hardy-Weinberg equilibrium (HWE) by using the Chi-square test. The association analysis, including co-dominant, dominant, recessive, over-dominant and log-additive models, was performed using SNPStats software [24]. The Akaike information criteria (AIC) was used to select the best fitted model. The model with the lowest AIC value was the best model. Odds ratios (ORs) with 95% confidence intervals (CIs) for each model were calculated using logistic regression. P- value $\leq 0.05$ was considered significant.

## Results

### Study population characteristics

A total of 233 infants (49.36% males) were enrolled into the study. There were 92 infants with confirmed cCMV infection (cCMV group) and 141 infants with excluded cCMV infection (healthy control group). The maternal and neonatal demographic data and general characteristics of two study groups at admission to Neonatal Department (NICU) are presented in Table 1. Infants with cCMV infection had statistically significant lower birth weight, head circumference, gestational age and younger mothers.

Newborn infants with confirmed cCMV infection (cCMV group) were symptomatic (n = 73) or asymptomatic (n = 19) at birth (Fig 1).

The clinical manifestation of symptomatic newborns is presented in Table 2. Microcephaly was observed in 23,9% of cases. All newborns with cCMV infection had brain US imaging, followed by brain MRI in over 95% of cases (in less than 5% of cases the MRI imaging was not conducted for technical reasons). Abnormal brain US was noted in 78.26% of cases, while abnormal brain MRI was noted in 86.21% of cases. Abnormalities of reticuloendothelial system such as petechiae, hepato- and splenomegaly occurred in 16.3%, 16.3%, 18.5% of infected newborns, respectively (Table 2).

### Results of SNP genotyping

Eight candidate SNPs were genotyped in 233 infants. All SNPs were in HWE (p>0.05) both in cCMV group and healthy control group.

### Associations between SNPs and cCMV infection

Eight selected SNPs were genotyped successfully in all study population. No statistically significant association was found between genotyped SNPs and cCMV infection (Table 3).

### Associations between SNPs and symptomatic course of cCMV infection

No statistically significant association was discovered between genotyped SNPs and symptomatic course of cCMV infection based on adopted criteria (Table 4).

**Table 1. Baseline characteristics of the study population including infants with cCMV and healthy control group.**

| Characteristics | Study population n = 233 | cCMV group n = 92 | Control group n = 141 | P-value[b] |
|---|---|---|---|---|
| Maternal age at delivery, years | 29 (26–33) | 28(25–31) | 29(26–33) | 0.004[a] |
| Gestational age, weeks | 39 (37–40) | 38(37–39) | 39(38–40) | 0.001[a] |
| Cesarean section, n (%) | 119 (59.17) | 49 (54.44) | 70 (54.69) | NS |
| Male, n (%) | 115 (49.36) | 46 (50.00) | 69 (48.94) | NS |
| Birth weight, g | 3080 (2590–3580) | 2755 (2170–3200) | 3270 (2860–3680) | 0.001[a] |
| IUGR (<10 percentile), n (%) | 42 (18.34) | 26 (28.26) | 16 (11.68) | 0.001[a] |
| Head circumference at admission to NICU, cm | 35.0 (33.5–36.5) | 34.1 (32.0–35.5) | 35.5 (34.5–37.0) | 0.001[a] |
| Length of hospitalization, days | 6 (3–10) | 8 (5–13) | 4 (2–9) | 0.001[a] |
| **Blood tests results:** | | | | |
| Thrombocytes, K/μl | 346 (214–476) | 187 (70–325) | 410 (319–501) | 0.001[a] |
| Neutrophils, K/μl | 2352 (1520–3616) | 1708 (1255–2676) | 3000 (1971–4154) | 0.001[a] |
| ASPAT, U/l | 37 (29–49) | 37 (29–52) | 37 (30–46) | NS |
| ALAT, U/l | 20 (14–28) | 20 (15–31) | 21 (14–27) | NS |
| GGT, U/l | 108 (66–186) | 123 (64–236) | 102 (66–162) | NS |
| Total bilirubin, mg/dl | 4.7 (1.4–8.4) | 2.3 (0.9–5.5) | 5.8 (2.5–9.3) | 0.001[a] |
| Direct bilirubin, mg/dl | 0.6 (0.5–0.8) | 0.6 (0.4–0.9) | 0.6 (0.5–0.8) | NS |

Data are presented as median (IQR, interquartile range) or number (%).

IUGR, intrauterine growth restriction; cCMV, congenital CMV infection; ASPAT, aspartate aminotransaminase; ALAT, alanine aminotransferase; GGT, Gamma-glutamyl transferase; NS, not significant (p-values above 0.05).

[a] P-value ≤ 0.05 are statistically significant.

[b] P-value for comparison between cCMV group and control group.

## Associations between selected gene SNPs and particular symptoms of cCMV infection

A summary of associations between examined SNPs and symptoms of cCMV infection is shown in Tables 5–8. SNP of IL12B rs3212227 was associated with decreased risk of prematurity (OR = 0.37, 95%CI, 0.14–0.98; p = 0.025) in log-additive model (Table 5). In relation to reticuloendothelial system abnormalities, IL1B rs19944 polymorphism was associated with the reduced risk of splenomegaly (OR = 0.36, 95%CI, 0.14–0.98; p = 0.034) in log-additive model (Table 6). Regarding laboratory blood results, heterozygous T/C genotype of IL28B rs12979860 polymorphism was associated with increased risk of thrombocytopenia (OR = 2.55; 95% CI, 1.03–6.32; p = 0.042) (Table 7), while infants carrying heterozygous T/C genotype at TLR4 rs4986791 had significantly increased risk of hepatitis (OR = 7.80; 95%CI, 1.49–40.81; p = 0.024) (Table 8). We did not find any other associations between examined SNPs and analyzed symptoms of cCMV group (S1–S9 Tables).

## Discussion

This is the first report on the relation between eights elected SNPs in genes encoding cytokines and cytokine receptors (IL1B rs16944, IL12B rs3212227, IL28B rs12979860, CCL2 rs1024611, DC-SIGN rs735240, TLR2rs5743708, TLR4 rs4986791, TLR9 rs352140), and the risk of cCMV infection and symptoms of the disease in the Caucasian newborns with cCMV infection. We found no association between analyzed SNPs and cCMV infection or symptomatic course of disease. However, our previous study documented, that rare T/T genotype of IL1B rs16944 polymorphism was linked to over two-fold higher risk of cCMV infection [13]. High IL-1β level was suggested to provoke an activation of Th1 (T-helper) cells and inflammatory

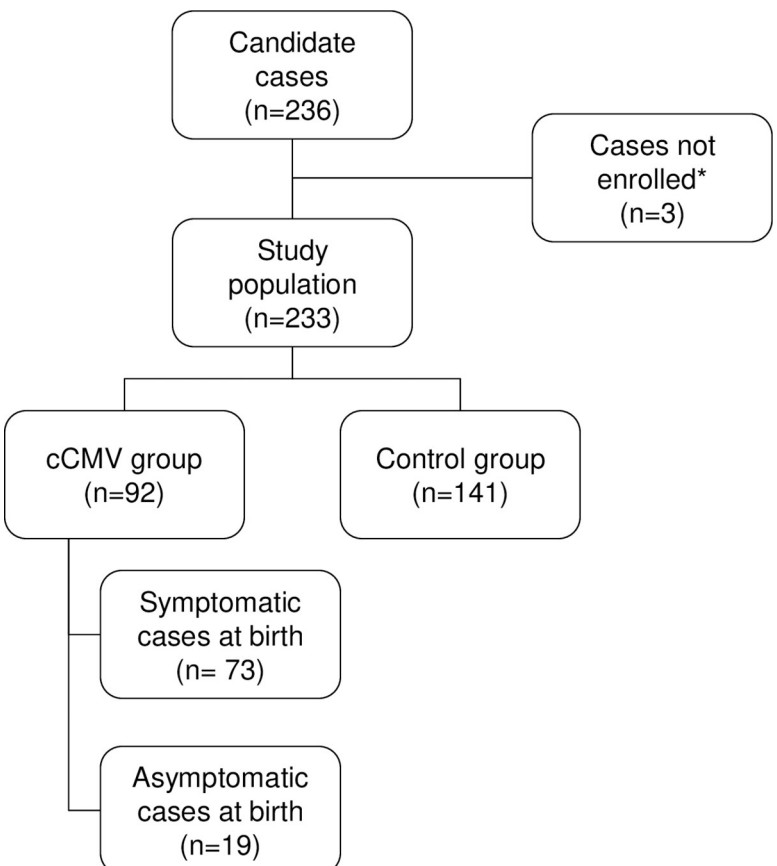

**Fig 1. Description of the study population.** Data are presented as number (n); cCMV–congenital CMV infection.
*Cases not enrolled to the study did not meet inclusion criteria due to lack of parental consent.

response [25]. We speculate that the lower sample size of the control group in the current study might explain different results [13]. Interestingly, Wujcicka et al found that heterozygotous C/T genotype of IL1A -899 polymorphism increased the risk of cCMV infection and development of symptomatic infection [22]. Moreover, some SNPs of Toll-like receptors (TLRs) were suggested to promote cCMV infection. It was reported that heterozygotous G/A genotype of TLR9 rs352140 polymorphism increased the risk of cCMV infection by 4.81 times, while heterozygotous G/A genotype of TLR2 rs5743708 polymorphism increased that risk by 10 times. However, we could not replicate these findings in our study. This discrepancy could be because of differences in the study population (due to different enrollment criteria). In addition, both previous studies were performed on merely about twenty cCMV-infected cases and similar number of cases of control group [10, 19].

To the best of our knowledge, there is limited data on associations between SNPs and particular symptoms in cCMV infection so far. Unexpectedly, we found no association between selected SNPs and CNS abnormalities or hearing loss in present study. Although, association between polymorphism of CCL2 rs1024611 and SNHL was previously demonstrated in a larger cohort [13]. Nevertheless, current study turned out to be the first one which proved novel associations between examined SNPs and particular symptoms involving hepatobiliary and reticuloendothelial system disorders and prematurity. Firstly, we observed a relation between decreased risk of prematurity and SNP polymorphism of IL12B rs3212227 in newborn infants with cCMV infection. It is well known that herpes virus infection may result in

**Table 2. Clinical symptoms and biochemical abnormalities in newborn infants with cCMV infection at diagnosis.**

| Characteristics | cCMV group n = 92 |
|---|---|
| **Central nervous system abnormalities** | |
| Microcephaly, n (%) | 22 (23.91) |
| Opisthotonus, n (%) | 11 (12.09) |
| Seizures, n (%) | 2 (2.17) |
| Abnormal muscle tone, n (%) | 53 (58.89) |
| Abnormal ultrasonography imaging, n (%) | 72 (78.26) |
| Abnormal magnetic resonance imaging, n (%) | 75 (86.21) (n = 87) |
| Chorioretinitis, n (%) | |
| Both eyes | 8 (8.70) |
| At least one eye | 17 (18.48) |
| Abnormal OEA, n (%) | |
| Right ear | 27 (29.35) |
| Left ear | 31 (33.70) |
| Both ears | 22 (23.91) |
| At least one ear | 36 (39.13) |
| Abnormal ABR, n (%) | |
| Right ear | 21 (25.30) (n = 82) |
| Left ear | 19 (23.45) (n = 81) |
| Both ears | 13 (16.05) |
| At least one ear | 29 (35.80) |
| **Hepatobiliary and reticuloendothelial abnormalities** | |
| Petechiae, n (%) | 15 (16.30) |
| Hepatomegaly, n (%) | 15 (16.30) |
| Splenomegaly, n (%) | 17 (18.48) |
| Thrombocytopenia ($<$100 000 platelets/mm$^3$), n (%) | 28 (30.43) |
| Neutropenia ($<$1000 neutrophils/µl), n (%) | 10 (10.87) |
| Anemia (relatively to the day of life), n (%) | 33 (35.87) |
| Elevated level of ASPAT ($>$84 U/l), n (%) | 8 (8.79) |
| Elevated level of ALAT ($>$60 U/l), n (%) | 8 (8.79) |
| Elevated level of GGT ($>$203 U/l), n (%) | 27 (32.53) |
| Cholestasis (direct bilirubin $>$1mg/dl), n (%) | 15 (17.44) |
| Hiperbilirubinemia (total bilirubin $>$12mg/dl), n (%) | 0 (0.00) |
| Incorrect coagulogram (INR $>$1.2; APTT $>$45.0 sec), n (%) | 1 (1.18) |

Data are presented as number (%).

OAE, Otoemission Acustic; ABR, Auditory Brainstem Response; Aspartate aminotransaminase, ASPAT; ALAT, Alanine aminotransferase; GGTP, Gamma-glutamyltransferase; INR, International Normalized Ratio; APTT, Activated Partial Thromboplastin Time.

pregnancy complications such as preterm birth [26]. Furthermore, CMV infection in preterm infants may promote more complications such as bronchopulmonary dysplasia (BPD) in comparison to term infants [27–31]. However, prematurity is not a pathognomonic symptom for cCMV infection. Considering limited data, this study was the first one that showed the link between IL12B rs3212227 polymorphism and reduced risk o prematurity in cCMV-infected patients. IL-12 is an inflammatory cytokine that may influence on Th1 cells, which take part in host immune defense against infection. So far, a number of previous studies have reported associations of IL-12 polymorphisms and other infectious diseases. It was hypothesized that

**Table 3. Associations between eight selected SNPs and cCMV infection.**

| Gene | dbSNP ID number[a] (alleles) | Genetic Model | Genotype | Control Group n = 141 | cCMV group n = 92 | OR (95% CI) | P-value[b] |
|---|---|---|---|---|---|---|---|
| IL1B | rs16944 (G/A) | Codominant | G/G | 71 (50.4) | 36 (39.1) | 1.00 | NS |
| | | | A/G | 57 (40.4) | 49 (53.3) | 1.70 (0.97–2.95) | |
| | | | A/A | 13 (9.2) | 7 (7.6) | 1.06 (0.39–2.89) | |
| | | Dominant | G/G | 71 (50.4) | 36 (39.1) | 1.00 | NS |
| | | | A/G-A/A | 70 (49.6) | 56 (60.9) | 1.58 (0.93–2.69) | |
| | | Recessive | G/G-A/G | 128 (90.8) | 85 (92.4) | 1.00 | NS |
| | | | A/A | 13 (9.2) | 7 (7.6) | 0.81 (0.31–2.12) | |
| | | Overdominant | G/G-A/A | 84 (59.6) | 43 (46.7) | 1.00 | NS |
| | | | A/G | 57 (40.4) | 49 (53.3) | 1.68 (0.99–2.85) | |
| | | Log-additive | - | - | - | 1.27 (0.84–1.91) | NS |
| IL12B | rs3212227 (G/T) | Codominant | T/T | 82 (58.2) | 57 (62) | 1.00 | NS |
| | | | T/G | 46 (32.6) | 28 (30.4) | 0.88 (0.49–1.56) | |
| | | | G/G | 13 (9.2) | 7 (7.6) | 0.77 (0.29–2.06) | |
| | | Dominant | T/T | 82 (58.2) | 57 (62.0) | 1.00 | NS |
| | | | T/G-G/G | 59 (41.8) | 35 (38.0) | 0.85 (0.50–1.46) | |
| | | Recessive | T/T-T/G | 128 (90.8) | 85 (92.4) | 1.00 | NS |
| | | | G/G | 13 (9.2) | 7 (7.6) | 0.81 (0.31–2.12) | |
| | | Overdominant | T/T-G/G | 95 (67.4) | 64 (69.6) | 1.00 | NS |
| | | | T/G | 46 (32.6) | 28 (30.4) | 0.90 (0.51–1.59) | |
| | | Log-additive | - | - | - | 0.88 (0.58–1.32) | NS |
| IL28B | rs12979860 (C/T) | Codominant | C/C | 65 (46.1) | 41 (44.6) | 1.00 | NS |
| | | | T/C | 59 (41.8) | 38 (41.3) | 1.02 (0.58–1.80) | |
| | | | T/T | 17 (12.1) | 13 (14.1) | 1.29 (0.53–2.76) | |
| | | Dominant | C/C | 65 (46.1) | 41 (44.6) | 1.00 | NS |
| | | | T/C-T/T | 76 (53.9) | 51 (55.4) | 1.06 (0.63–1.80) | |
| | | Recessive | C/C-T/C | 124 (87.9) | 79 (85.9) | 1.00 | NS |
| | | | T/T | 17 (12.1) | 13 (14.1) | 1.20 (0.55–2.61) | |
| | | Overdominant | C/C-T/T | 82 (58.2) | 54 (58.7) | 1.00 | NS |
| | | | T/C | 59 (41.8) | 38 (41.3) | 0.98 (0.57–1.67) | |
| | | Log-additive | - | - | - | 1.08 (0.74–1.58) | NS |
| CCL2 | rs1024611 (A/G) | Codominant | A/A | 74 (52.5) | 50 (54.4) | 1.00 | NS |
| | | | G/A | 57 (40.4) | 39 (42.4) | 1.01 (0.59–1.74) | |
| | | | G/G | 10 (7.1) | 3 (3.3) | 0.44 (0.12–1.69) | |
| | | Dominant | A/A | 74 (52.5) | 50 (54.4) | 1.00 | NS |
| | | | G/A-G/G | 67 (47.5) | 42 (45.6) | 0.93 (0.55–1.57) | |
| | | Recessive | A/A-G/A | 131 (92.9) | 89 (96.7) | 1.00 | NS |
| | | | G/G | 10 (7.1) | 3 (3.3) | 0.44 (0.12–1.65) | |
| | | Overdominant | A/A-G/G | 84 (59.6) | 53 (57.6) | 1.00 | NS |
| | | | G/A | 57 (40.4) | 39 (42.4) | 1.08 (0.64–1.85) | |
| | | Log-additive | - | - | - | 0.85 (0.55–1.33) | NS |
| DC-SIGN | rs735240 (A/G) | Codominant | G/G | 53 (37.6) | 35 (38.0) | 1.00 | NS |
| | | | G/A | 65 (46.1) | 37 (40.2) | 0.86 (0.48–1.55) | |
| | | | A/A | 23 (16.3) | 20 (21.7) | 1.32 (0.63–2.75) | |
| | | Dominant | G/G | 53 (37.6) | 35 (38.0) | 1.00 | NS |
| | | | G/A-A/A | 88 (62.4) | 57 (62.0) | 0.98 (0.57–1.69) | |
| | | Recessive | G/G-G/A | 118 (83.7) | 72 (78.3) | 1.00 | NS |
| | | | A/A | 23 (16.3) | 20 (21.7) | 1.43 (0.73–2.78) | |
| | | Overdominant | G/G-A/A | 76 (53.9) | 55 (59.8) | 1.00 | NS |
| | | | G/A | 6 (46.1) | 37 (40.2) | 0.79 (0.46–1.34) | |
| | | Log-additive | - | - | - | 1.10 (0.77–1.58) | NS |
| TLR2 | rs5743708 (A/G) | - | G/G | 126 (89.4) | 82 (89.1) | 1.00 | NS |
| | | | G/A | 15 (10.6) | 10 (10.9) | 1.02 (0.44–2.39) | |

*(Continued)*

**Table 3.** (Continued)

| Gene | dbSNP ID number[a] (alleles) | Genetic Model | Genotype | Control Group n = 141 | cCMV group n = 92 | OR (95% CI) | P-value[b] |
|------|------------------------------|---------------|----------|-----------------------|-------------------|-------------|------------|
| TLR4 | rs4986791 (C/T) | Codominant | C/C | 126 (89.4) | 83 (90.2) | 1.00 | NS |
| | | | T/C | 13 (9.2) | 9 (9.8) | 1.05 (0.43–2.57) | |
| | | | T/T | 2 (1.4) | 0 (0.0) | 0.00 (0.00-NA) | |
| | | Dominant | C/C | 126 (89.4) | 83 (90.2) | 1.00 | NS |
| | | | T/C-T/T | 15 (10.6) | 9 (9.8) | 0.91 (0.38–2.18) | |
| | | Recessive | C/C-T/C | 139 (98.6) | 92 (100.0) | 1.00 | NS |
| | | | T/T | 2 (1.4) | 0 (0.0) | 0.00 (0.00-NA) | |
| | | Overdominant | C/C-T/T | 128 (90.8) | 83 (90.2) | 1.00 | NS |
| | | | T/C | 13 (9.2) | 9 (9.8) | 1.07 (0.44–2.61) | |
| | | Log-additive | - | - | - | 0.82 (0.37–1.81) | NS |
| TLR | rs352140 (C/T) | Codominant | T/T | 44 (31.2) | 30 (32.6) | 1.00 | NS |
| | | | T/C | 74 (52.5) | 47 (51.1) | 0.93 (0.52–1.68) | |
| | | | C/C | 23 (16.3) | 15 (16.3) | 0.96 (0.43–2.13) | |
| | | Dominant | T/T | 44 (31.2) | 30 (32.6) | 1.00 | NS |
| | | | T/C-C/C | 97 (68.8) | 62 (67.4) | 0.94 (0.53–1.65) | |
| | | Recessive | T/T-T/C | 118 (83.7) | 77 (83.7) | 1.00 | NS |
| | | | C/C | 23 (16.3) | 15 (16.3) | 1.00 (0.49–2.03) | |
| | | Overdominant | T/T-C/C | 67 (47.5) | 45 (48.9) | 1.00 | NS |
| | | | T/C | 74 (52.5) | 47 (51.1) | 0.95 (0.56–1.60) | |
| | | Log-additive | - | - | - | 0.97 (0.66–1.43) | NS |

Data presented as number (%), OR, odds ratio; CI, confidence interval; NA, not applicable; NS, not significant (p-values above 0.05); IL, Interleukin; CCL 2, C-C motif chemokine ligand 2; DC-SIGN, dendritic cell-specific ICAM-grabbing non-integrin; TLR, Toll-like receptor.

[a] SNP database (dbSNP) reference number (ID number).

[b] P-value for comparison between control group and cCMV group.

**Table 4. Associations between selected SNPs and symptomatic or asymptomatic course of cCMV infection.**

| Gene | dbSNP ID number[a] (alleles) | Genetic model | Genotype | Asymptomatic n = 19 | Symptomatic n = 73 | OR (95% CI) | P-value[b] |
|------|------------------------------|---------------|----------|---------------------|--------------------|-------------|------------|
| IL1B | rs16944 (G/A) | Codominant | G/G | 7 (36.8) | 29 (39.7) | 1.00 | NS |
| | | | A/G | 10 (52.6) | 39 (53.4) | 0.94 (0.32–2.77) | |
| | | | A/A | 2 (10.5) | 5 (6.8) | 0.60 (0.10–3.78) | |
| | | Dominant | G/G | 7 (36.8) | 29 (39.7) | 1.00 | NS |
| | | | A/G-A/A | 12 (63.2) | 44 (60.3) | 0.89 (0.31–2.51) | |
| | | Recessive | G/G-A/G | 17 (89.5) | 68 (93.2) | 1.00 | NS |
| | | | A/A | 2 (10.5) | 5 (6.8) | 0.62 (0.11–3.50) | |
| | | Overdominant | G/G-A/A | 9 (47.4) | 34 (46.6) | 1 | |
| | | | A/G | 10 (52.6) | 39 (53.4) | 1.03 (0.38–2.84) | NS |
| | | Log-additive | - | - | - | 0.84 (0.37–1.92) | NS |
| IL12B | rs3212227 (G/T) | Codominant | T/T | 14 (73.7) | 43 (58.9) | 1.0 | NS |
| | | | T/G | 4 (21.1) | 24 (32.9) | 1.95 (0.58–6.61) | |
| | | | G/G | 1 (5.3) | 6 (8.2) | 1.95 (0.22–17.65) | |
| | | Dominant | T/T | 14 (73.7) | 43 (58.9) | 1.00 | NS |
| | | | T/G-G/G | 5 (26.3) | 30 (41.1) | 1.95 (0.64–6.00) | |
| | | Recessive | T/T-T/G | 18 (94.7) | 67 (91.8) | 1.00 | NS |
| | | | G/G | 1 (5.3) | 6 (8.2) | 1.61 (0.18–14.26) | |
| | | Overdominant | T/T-G/G | 15 (79.0) | 49 (67.1) | 1.00 | NS |
| | | | T/G | 4 (21.1) | 24 (32.9) | 1.84 (0.55–6.14) | |
| | | Log-additive | - | - | - | 1.64 (0.66–4.07) | NS |

*(Continued)*

**Table 4.** (Continued)

| Gene | dbSNP ID number[a] (alleles) | Genetic model | Genotype | Asymptomatic n = 19 | Symptomatic n = 73 | OR (95% CI) | P-value[b] |
|---|---|---|---|---|---|---|---|
| IL28B | rs12979860 (C/T) | Codominant | C/C | 11 (57.9) | 30 (41.1) | 1.00 | NS |
| | | | T/C | 6 (31.6) | 32 (43.8) | 1.96 (0.64–5.95) | |
| | | | T/T | 2 (10.5) | 11 (15.1) | 2.02 (0.38–10.58) | |
| | | Dominant | C/C | 11 (57.9) | 30 (41.1) | 1.00 | NS |
| | | | T/C-T/T | 8 (42.1) | 43 (58.9) | 1.97 (0.71–5.48) | |
| | | Recessive | C/C-T/C | 17 (89.5) | 62 (84.9) | 1.00 | NS |
| | | | T/T | 2 (10.5) | 11 (15.1) | 1.51 (0.30–7.46) | |
| | | Overdominant | C/C-T/T | 13 (68.4) | 41 (56.2) | 1.0 | NS |
| | | | T/C | 6 (31.6) | 32 (43.8) | 1.69 (0.58–4.94) | |
| | | Log-additive | - | - | - | 1.59 (0.73–3.42) | NS |
| CCL2 | rs1024611 (A/G) | Codominant | A/A | 10 (52.6) | 40 (54.8) | 1.00 | NS |
| | | | G/A | 9 (47.4) | 30 (41.1) | 0.83 (0.30–2.30) | |
| | | | G/G | 0 (0.0) | 3 (4.1) | NA (0.00-NA) | |
| | | Dominant | A/A | 10 (52.6) | 40 (54.8) | 1.0 | NS |
| | | | G/A-G/G | 9 (47.4) | 33 (45.2) | 0.92 (0.33–2.52) | |
| | | Recessive | A/A-G/A | 19 (100.0) | 70 (95.9) | 1.00 | NS |
| | | | G/G | 0 (0.0) | 3 (4.1) | NA (0.00-NA) | |
| | | Overdominant | A/A-G/G | 10 (52.6) | 43 (58.9) | 1.00 | NS |
| | | | G/A | 9 (47.4) | 30 (41.1) | 0.78 (0.28–2.14) | |
| | | Log-additive | - | - | - | 1.06 (0.43–2.63) | NS |
| DC-SIGN | rs735240 (A/G) | Codominant | G/G | 7 (36.8) | 28 (38.4) | 1.00 | NS |
| | | | G/A | 6 (31.6) | 31 (42.5) | 1.29 (0.39–4.31) | |
| | | | A/A | 6 (31.6) | 14 (19.2) | 0.58 (0.16–2.07) | |
| | | Dominant | G/G | 7 (36.8) | 28 (38.4) | 1.00 | NS |
| | | | G/A-A/A | 12 (63.2) | 45 (61.6) | 0.94 (0.33–2.67) | |
| | | Recessive | G/G-G/A | 13 (68.4) | 59 (80.8) | 1.00 | NS |
| | | | A/A | 6 (31.6) | 14 (19.2) | 0.51 (0.17–1.59) | |
| | | Overdominant | G/G-A/A | 13 (68.4) | 42 (57.5) | 1.00 | NS |
| | | | G/A | 6 (31.6) | 31 (42.5) | 1.60 (0.55–4.68) | |
| | | Log-additive | - | - | - | 0.79 (0.40–1.53) | NS |
| TLR2 | rs5743708 (A/G) | - | G/G | 17 (89.5) | 65 (89.0) | 1.00 | NS |
| | | | G/A | 2 (10.5) | 8 (11.0) | 1.05 (0.20–5.39) | |
| TLR4 | rs4986791 (C/T) | - | C/C | 17 (89.5) | 66 (90.4) | 1.00 | NS |
| | | | T/C | 2 (10.5) | 7 (9.6) | 0.90 (0.17–4.74) | |
| TLR9 | rs352140 (C/T) | Codominant | T/T | 7 (36.8) | 23 (31.5) | 1.00 | NS |
| | | | T/C | 9 (47.4) | 38 (52.0) | 1.29 (0.42–3.92) | |
| | | | C/C | 3 (15.8) | 12 (16.4) | 1.22 (0.27–5.58) | |
| | | Dominant | T/T | 7 (36.8) | 23 (31.5) | 1.00 | NS |
| | | | T/C-C/C | 12 (63.2) | 50 (68.5) | 1.27 (0.44–3.64) | |
| | | Recessive | T/T-T/C | 16 (84.2) | 61 (83.6) | 1.00 | NS |
| | | | C/C | 3 (15.8) | 12 (16.4) | 1.05 (0.26–4.17) | |
| | | Overdominant | T/T-C/C | 10 (52.6) | 35 (48.0) | 1.00 | NS |
| | | | T/C | 9 (47.4) | 38 (52.0) | 1.21 (0.44–3.32) | |
| | | Log-additive | - | - | - | 1.14 (0.54–2.41) | NS |

Data presented as number (%), OR, odds ratio; CI, confidence interval; NA, not applicable; NS, not significant (p-values above 0.05); IL, Interleukin; CCL 2, C-C motif chemokine ligand 2; DC-SIGN dendritic cell-specific ICAM-grabbing non-integrin; TLR, Toll-like receptor.

[a] SNP database (dbSNP) reference number (ID number).

[b] P-value for comparison between asymptomatic and symptomatic infants in cCMV group.

**Table 5. Associations between SNPs and prematurity in newborn infants with cCMV infection.**

| Gene | dbSNP ID number[a] (alleles) | Genetic Model | Genotype | Term infants n = 69 | Pre-term infants n = 23 | OR (95% CI) | P-value[b] | AIC |
|---|---|---|---|---|---|---|---|---|
| IL1B | rs16944 (G/T) | Codominant | G/G | 25 (36.2) | 11 (47.8) | 1.00 | 0.1 | 104.9 |
| | | | A/G | 37 (53.6) | 12 (52.2) | 0.74 (0.28–1.93) | | |
| | | | A/A | 7 (10.1) | 0 (0.0) | 0.00 (0.00-NA) | | |
| | | Dominant | G/G | 25 (36.2) | 11 (47.8) | 1.00 | 0.33 | 106.5 |
| | | | A/G-A/A | 44 (63.8) | 12 (52.2) | 0.62 (0.24–1.61) | | |
| | | Recessive | G/G-A/G | 62 (89.9) | 23 (100.0) | 1.00 | 0.04 | 103.3 |
| | | | A/A | 7 (10.1) | 0 (0.0) | 0.00 (0.00-NA) | | |
| | | Overdominant | G/G-A/A | 32 (46.4) | 11 (47.8) | 1.00 | 0.9 | 107.5 |
| | | | A/G | 37 (53.6) | 12 (52.2) | 0.94 (0.37–2.43) | | |
| | | Log-additive | - | - | - | 0.54 (0.24–1.23) | 0.13 | 105.2 |
| IL12B | rs3212227 (G/T) | Codominant | T/T | 39 (56.5) | 18 (78.3) | 1.00 | 0.047 | 103.4 |
| | | | T/G | 23 (33.3) | 5 (21.7) | 0.47 (0.15–1.44) | | |
| | | | G/G | 7 (10.1) | 0 (0.0) | 0.00 (0.00-NA) | | |
| | | Dominant | T/T | 39 (56.5) | 18 (78.3) | 1.00 | 0.056 | 103.8 |
| | | | T/G-G/G | 30 (43.5) | 5 (21.7) | 0.36 (0.12–1.08) | | |
| | | Recessive | T/T-T/G | 62 (89.9) | 23 (100.0) | 1.00 | 0.040 | 103.3 |
| | | | G/G | 7 (10.1) | 0 (0.0) | 0.00 (0.00-NA) | | |
| | | Overdominant | T/T-G/G | 46 (66.7) | 18 (78.3) | 1.00 | 0.28 | 106.3 |
| | | | T/G | 23 (33.3) | 5 (21.7) | 0.56 (0.18–1.69) | | |
| | | Log-additive | - | - | - | 0.37 (0.14–0.98) | 0.025[c] | 102.4 |
| IL28B | rs12979860 (C/T) | Codominant | C/C | 32 (46.4) | 9 (39.1) | 1.00 | 0.2 | 106.2 |
| | | | T/C | 30 (43.5) | 8 (34.8) | 0.95 (0.32–2.78) | | |
| | | | T/T | 7 (10.1) | 6 (26.1) | 3.05 (0.82–11.38) | | |
| | | Dominant | C/C | 32 (46.4) | 9 (39.1) | 1.00 | 0.54 | 107.1 |
| | | | T/C-T/T | 37 (53.6) | 14 (60.9) | 1.35 (0.51–3.52) | | |
| | | Recessive | C/C-T/C | 62 (89.9) | 17 (73.9) | 1.00 | 0.072 | 104.2 |
| | | | T/T | 7 (10.1) | 6 (26.1) | 3.13 (0.93–10.54) | | |
| | | Overdominant | C/C-T/T | 39 (56.5) | 15 (65.2) | 1.00 | 0.46 | 106.9 |
| | | | T/C | 30 (43.5) | 8 (34.8) | 0.69 (0.26–1.85) | | |
| | | Log-additive | - | - | - | 1.58 (0.82–3.07) | 0.17 | 105.6 |
| CCL2 | rs1024611 (A/G) | Codominant | A/A | 40 (58.0) | 10 (43.5) | 1.00 | 0.48 | 108 |
| | | | G/A | 27 (39.1) | 12 (52.2) | 1.78 (0.67–4.69) | | |
| | | | G/G | 2 (2.9) | 1 (4.3) | 2.00 (0.16–24.33) | | |
| | | Dominant | A/A | 40 (58.0) | 10 (43.5) | 1.00 | 0.23 | 106 |
| | | | G/A-G/G | 29 (42.0) | 13 (56.5) | 1.79 (0.69–4.65) | | |
| | | Recessive | A/A-G/A | 67 (97.1) | 22 (95.7) | 1.00 | 0.74 | 107.4 |
| | | | G/G | 2 (2.9) | 1 (4.3) | 1.52 (0.13–17.62) | | |
| | | Overdominant | A/A-G/G | 42 (60.9) | 11 (47.8) | 1.00 | 0.28 | 106.3 |
| | | | G/A | 27 (39.1) | 12 (52.2) | 1.70 (0.66–4.39) | | |
| | | Log-additive | - | - | - | 1.64 (0.72–3.74) | 0.24 | 106.1 |
| DC-SIGN | rs735240 (A/G) | Codominant | G/G | 25 (36.2) | 10 (43.5) | 1.00 | 0.79 | 109 |
| | | | G/A | 29 (42.0) | 8 (34.8) | 0.69 (0.24–2.02) | | |
| | | | A/A | 15 (21.7) | 5 (21.7) | 0.83 (0.24–2.91) | | |
| | | Dominant | G/G | 25 (36.2) | 10 (43.5) | 1.00 | 0.54 | 107.1 |
| | | | G/A-A/A | 44 (63.8) | 13 (56.5) | 0.74 (0.28–1.93) | | |
| | | Recessive | G/G-G/A | 54 (78.3) | 18 (78.3) | 1.00 | 1 | 107.5 |
| | | | A/A | 15 (21.7) | 5 (21.7) | 1.00 (0.32–3.14) | | |
| | | Overdominant | G/G-A/A | 40 (58.0) | 15 (65.2) | 1.00 | 0.54 | 107.1 |
| | | | G/A | 29 (42.0) | 8 (34.8) | 0.74 (0.28–1.96) | | |
| | | Log-additive | - | - | - | 0.88 (0.47–1.65) | 0.69 | 107.3 |
| TLR2 | rs5743708 (A/G) | - | G/G | 60 (87.0) | 22 (95.7) | 1.00 | 0.21 | 105.9 |
| | | | G/A | 9 (13.0) | 1 (4.3) | 0.30 (0.04–2.53) | | |

(*Continued*)

**Table 5.** (Continued)

| Gene | dbSNP ID number[a] (alleles) | Genetic Model | Genotype | Term infants n = 69 | Pre-term infants n = 23 | OR (95% CI) | P-value[b] | AIC |
|------|------------------------------|---------------|----------|---------------------|-------------------------|-------------|------------|-----|
| TLR4 | rs4986791 (C/T) | - | C/C | 64 (92.8) | 19 (82.6) | 1.00 | 0.18 | 105.7 |
| | | | T/C | 5 (7.2) | 4 (17.4) | 2.69 (0.66–11.05) | | |
| TLR9 | rs352140 (C/T) | Codominant | T/T | 26 (37.7) | 4 (17.4) | 1.00 | 0.17 | 105.9 |
| | | | T/C | 33 (47.8) | 14 (60.9) | 2.76 (0.81–9.38) | | |
| | | | C/C | 10 (14.5) | 5 (21.7) | 3.25 (0.72–14.62) | | |
| | | Dominant | T/T | 26 (37.7) | 4 (17.4) | 1.00 | 0.062 | 104 |
| | | | T/C-C/C | 43 (62.3) | 19 (82.6) | 2.87 (0.88–9.38) | | |
| | | Recessive | T/T-T/C | 59 (85.5) | 18 (78.3) | 1.00 | 0.43 | 106.8 |
| | | | C/C | 10 (14.5) | 5 (21.7) | 1.64 (0.50–5.42) | | |
| | | Overdominant | T/T-C/C | 36 (52.2) | 9 (39.1) | 1.00 | 0.28 | 106.3 |
| | | | T/C | 33 (47.8) | 14 (60.9) | 1.70 (0.65–4.44) | | |
| | | Log-additive | - | - | - | 1.82 (0.90–3.70) | 0.093 | 104.6 |

Data presented as number (%), OR, odds ratio; CI, confidence interval; NA, not applicable; AIC, Akaike information criterion; p-values below 0.05 are statistically significant; IL, Interleukin; CCL 2, C-C motif chemokine ligand 2; DC-SIGN, dendritic cell-specific ICAM-grabbing non-integrin; TLR, Toll-like receptor.

[a] SNP database (dbSNP) reference number (ID number).

[b] P-value for comparison between term infants and pre-term infants in cCMV group.

[c] The best fitted model based on AIC.

IL12Brs3212227 polymorphism might reduce risk of invasive aspergillosis among immuno-compromised patients due to alternative splicing of IL12B mRNA and altering mRNA expression which in turn, might dysregulate IL12-mediated Th1 cells response [32]. The polymorphisms in IL12 gene were also proved to support host's defense against HCV infection through more efficient response to antiviral therapy. However, not only protective role of IL12B rs3212227 was previously observed. Some reports documented that several SNPs of

**Table 6. Associations between SNPs and splenomegaly in newborn infants with cCMV infection.**

| Gene | dbSNP ID number[a] (alleles) | Genetic Model | Genotype | Without splenomegaly n = 75 | With splenomegaly n = 17 | OR (95% CI) | P-value[b] | AIC |
|------|------------------------------|---------------|----------|-----------------------------|--------------------------|-------------|------------|-----|
| IL1B | rs16944 (G/A) | Codominant | G/G | 26 (34.7) | 10 (58.8) | 1.00 | 0.07 | 88.7 |
| | | | A/G | 42 (56.0) | 7 (41.2) | 0.43 (0.15–1.28) | | |
| | | | A/A | 7 (9.3) | 0 (0.0) | 0.00 (0.00-NA) | | |
| | | Dominant | G/G | 26 (34.7) | 10 (58.8) | 1.00 | 0.068 | 88.7 |
| | | | A/G-A/A | 49 (65.3) | 7 (41.2) | 0.37 (0.13–1.09) | | |
| | | Recessive | G/G-A/G | 68 (90.7) | 17 (100.0) | 1.00 | 0.084 | 89.1 |
| | | | A/A | 7 (9.3) | 0 (0.0) | 0.00 (0.00-NA) | | |
| | | Overdominant | G/G-A/A | 33 (44.0) | 10 (58.8) | 1.00 | 0.27 | 90.8 |
| | | | A/G | 42 (56.0) | 7 (41.2) | 0.55 (0.19–1.60) | | |
| | | Log-additive | - | - | - | 0.36 (0.14–0.98) | 0.034[c] | 87.6 |
| IL12B | rs3212227 (G/T) | Codominant | T/T | 45 (60.0) | 12 (70.6) | 1.00 | 0.21 | 90.9 |
| | | | T/G | 23 (30.7) | 5 (29.4) | 0.82 (0.26–2.60) | | |
| | | | G/G | 7 (9.3) | 0 (0.0) | 0.00 (0.00-NA) | | |
| | | Dominant | T/T | 45 (60.0) | 12 (70.6) | 1.00 | 0.41 | 91.4 |
| | | | T/G-G/G | 30 (40.0) | 5 (29.4) | 0.62 (0.20–1.96) | | |
| | | Recessive | T/T-T/G | 68 (90.7) | 17 (100.0) | 1.00 | 0.084 | 89.1 |
| | | | G/G | 7 (9.3) | 0 (0.0) | 0.00 (0.00-NA) | | |
| | | Overdominant | T/T-G/G | 52 (69.3) | 12 (70.6) | 1.00 | 0.92 | 92 |
| | | | T/G | 23 (30.7) | 5 (29.4) | 0.94 (0.30–2.98) | | |
| | | Log-additive | - | - | - | 0.56 (0.21–1.49) | 0.22 | 90.6 |

*(Continued)*

**Table 6.** (Continued)

| Gene | dbSNP ID number[a] (alleles) | Genetic Model | Genotype | Without splenomegaly n = 75 | With splenomegaly n = 17 | OR (95% CI) | P-value[b] | AIC |
|---|---|---|---|---|---|---|---|---|
| IL28B | rs12979860 (C/T) | Codominant | C/C | 34 (45.3) | 7 (41.2) | 1.00 | 0.86 | 93.8 |
| | | | T/C | 30 (40.0) | 8 (47.1) | 1.30 (0.42–4.00) | | |
| | | | T/T | 11 (14.7) | 2 (11.8) | 0.88 (0.16–4.89) | | |
| | | Dominant | C/C | 34 (45.3) | 7 (41.2) | 1.00 | 0.76 | 92 |
| | | | T/C-T/T | 41 (54.7) | 10 (58.8) | 1.18 (0.41–3.45) | | |
| | | Recessive | C/C-T/C | 64 (85.3) | 15 (88.2) | 1.00 | 0.75 | 92 |
| | | | T/T | 11 (14.7) | 2 (11.8) | 0.78 (0.16–3.87) | | |
| | | Overdominant | C/C-T/T | 45 (60.0) | 9 (52.9) | 1.00 | 0.6 | 91.8 |
| | | | T/C | 30 (40.0) | 8 (47.1) | 1.33 (0.46–3.84) | | |
| | | Log-additive | - | - | - | 1.03 (0.49–2.16) | 0.95 | 92.1 |
| CCL2 | rs1024611 (A/G) | Codominant | A/A | 41 (54.7) | 9 (52.9) | 1.00 | 0.82 | 93.7 |
| | | | G/A | 32 (42.7) | 7 (41.2) | 1.00 (0.33–2.97) | | |
| | | | G/G | 2 (2.7) | 1 (5.9) | 2.28 (0.19–27.93) | | |
| | | Dominant | A/A | 41 (54.7) | 9 (52.9) | 1.00 | 0.9 | 92 |
| | | | G/A-G/G | 34 (45.3) | 8 (47.1) | 1.07 (0.37–3.08) | | |
| | | Recessive | A/A-G/A | 73 (97.3) | 16 (94.1) | 1.00 | 0.53 | 91.7 |
| | | | G/G | 2 (2.7) | 1 (5.9) | 2.28 (0.19–26.72) | | |
| | | Overdominant | A/A-G/G | 43 (57.3) | 10 (58.8) | 1.00 | 0.91 | 92 |
| | | | G/A | 32 (42.7) | 7 (41.2) | 0.94 (0.32–2.74) | | |
| | | Log-additive | - | - | - | 1.17 (0.46–2.94) | 0.74 | 92 |
| DC-SIGN | rs735240 (A/G) | Codominant | G/G | 30 (40.0) | 5 (29.4) | 1.00 | 0.71 | 93.4 |
| | | | G/A | 29 (38.7) | 8 (47.1) | 1.66 (0.48–5.65) | | |
| | | | A/A | 16 (21.3) | 4 (23.5) | 1.50 (0.35–6.38) | | |
| | | Dominant | G/G | 30 (40.0) | 5 (29.4) | 1.00 | 0.41 | 91.4 |
| | | | G/A-A/A | 45 (60.0) | 12 (70.6) | 1.60 (0.51–5.01) | | |
| | | Recessive | G/G-G/A | 59 (78.7) | 13 (76.5) | 1.00 | 0.84 | 92 |
| | | | A/A | 16 (21.3) | 4 (23.5) | 1.13 (0.33–3.96) | | |
| | | Overdominant | G/G-A/A | 46 (61.3) | 9 (52.9) | 1.00 | 0.53 | 91.7 |
| | | | G/A | 29 (38.7) | 8 (47.1) | 1.41 (0.49–4.07) | | |
| | | Log-additive | - | - | - | 1.25 (0.63–2.49) | 0.53 | 91.7 |
| TLR2 | rs5743708 (A/G) | - | G/G | 67 (89.3) | 15 (88.2) | 1.00 | 0.9 | 92 |
| | | | G/A | 8 (10.7) | 2 (11.8) | 1.12 (0.21–5.80) | | |
| TLR4 | rs4986791 (C/T) | - | C/C | 70 (93.3) | 13 (76.5) | 1.00 | 0.056 | 88.4 |
| | | | T/C | 5 (6.7) | 4 (23.5) | 4.31 (1.02–18.22) | | |
| TLR9 | rs352140 (C/T) | Codominant | T/T | 24 (32.0) | 6 (35.3) | 1.00 | 0.93 | 93.9 |
| | | | T/C | 39 (52.0) | 8 (47.1) | 0.82 (0.25–2.66) | | |
| | | | C/C | 12 (16.0) | 3 (17.6) | 1.00 (0.21–4.71) | | |
| | | Dominant | T/T | 24 (32.0) | 6 (35.3) | 1.00 | 0.79 | 92 |
| | | | T/C-C/C | 51 (68.0) | 11 (64.7) | 0.86 (0.29–2.61) | | |
| | | Recessive | T/T-T/C | 63 (84.0) | 14 (82.3) | 1.00 | 0.87 | 92 |
| | | | C/C | 12 (16.0) | 3 (17.6) | 1.13 (0.28–4.52) | | |
| | | Overdominant | T/T-C/C | 36 (48.0) | 9 (52.9) | 1.00 | 0.71 | 91.9 |
| | | | T/C | 39 (52.0) | 8 (47.1) | 0.82 (0.29–2.36) | | |
| | | Log-additive | - | - | - | 0.96 (0.44–2.10) | 0.93 | 92 |

Data presented as number (%), OR, odds ratio; CI, confidence interval; NA, not applicable; AIC, Akaike information criterion; p-values below 0.05 are statistically significant; IL, Interleukin; CCL 2, C-C motif chemokine ligand 2; DC-SIGN, dendritic cell-specific ICAM-grabbing non-integrin; TLR, Toll-like receptor.

[a] SNP database (dbSNP) reference number (ID number).

[b] P-value for comparison between infants without splenomegaly and with splenomegaly in cCMV group.

[c] The best fitted model based on AIC.

**Table 7. Associations between SNPs and thrombocytopenia in newborn infants with cCMV infection.**

| Gene | dbSNP ID number[a] | Genetic Model | Genotype | Without thrombocytopenia n = 64 | With thrombocytopenia n = 28 | OR (95% CI) | P-value[b] | AIC |
|---|---|---|---|---|---|---|---|---|
| IL1B | rs16944 (G/A) | Codominant | G/G | 24 (37.5) | 12 (42.9) | 1.00 | 0.57 | 117.9 |
| | | | A/G | 34 (53.1) | 15 (53.6) | 0.88 (0.35–2.22) | | |
| | | | A/A | 6 (9.4) | 1 (3.6) | 0.33 (0.04–3.09) | | |
| | | Dominant | G/G | 24 (37.5) | 12 (42.9) | 1.00 | 0.63 | 116.8 |
| | | | A/G-A/A | 40 (62.5) | 16 (57.1) | 0.80 (0.32–1.97) | | |
| | | Recessive | G/G-A/G | 58 (90.6) | 27 (96.4) | 1.00 | 0.3 | 116 |
| | | | A/A | 6 (9.4) | 1 (3.6) | 0.36 (0.04–3.12) | | |
| | | Overdominant | G/G-A/A | 30 (46.9) | 13 (46.4) | 1.00 | 0.97 | 117.1 |
| | | | A/G | 34 (53.1) | 15 (53.6) | 1.02 (0.42–2.48) | | |
| | | Log-additive | - | - | - | 0.73 (0.35–1.55) | 0.41 | 116.4 |
| IL12B | rs3212227 (G/T) | Codominant | T/T | 37 (57.8) | 20 (71.4) | 1.00 | 0.37 | 117.1 |
| | | | T/G | 21 (32.8) | 7 (25) | 0.62 (0.22–1.70) | | |
| | | | G/G | 6 (9.4) | 1 (3.6) | 0.31 (0.03–2.74) | | |
| | | Dominant | T/T | 37 (57.8) | 20 (71.4) | 1.00 | 0.21 | 115.5 |
| | | | T/G-G/G | 27 (42.2) | 8 (28.6) | 0.55 (0.21–1.43) | | |
| | | Recessive | T/T-T/G | 58 (90.6) | 27 (96.4) | 1.00 | 0.3 | 116 |
| | | | G/G | 6 (9.4) | 1 (3.6) | 0.36 (0.04–3.12) | | |
| | | Overdominant | T/T-G/G | 43 (67.2) | 21 (75.0) | 1.00 | 0.45 | 116.5 |
| | | | T/G | 21 (32.8) | 7 (25.0) | 0.68 (0.25–1.86) | | |
| | | Log-additive | - | - | - | 0.59 (0.27–1.28) | 0.16 | 115.1 |
| IL28B | rs12979860 (C/T) | Codominant | C/C | 31 (48.4) | 10 (35.7) | 1.0 | 0.099 | 114.4 |
| | | | T/C | 22 (34.4) | 16 (57.1) | 2.25 (0.86–5.89) | | |
| | | | T/T | 11 (17.2) | 2 (7.1) | 0.56 (0.11–2.98) | | |
| | | Dominant | C/C | 31 (48.4) | 10 (35.7) | 1.0 | 0.26 | 115.8 |
| | | | T/C-T/T | 33 (51.6) | 18 (64.3) | 1.69 (0.68–4.22) | | |
| | | Recessive | C/C-T/C | 53 (82.8) | 26 (92.9) | 1.0 | 0.18 | 115.3 |
| | | | T/T | 11 (17.2) | 2(7.1) | 0.37 (0.08–1.80) | | |
| | | Overdominant | C/C-T/T | 42 (65.6) | 12 (42.9) | 1.00 | 0.042[c] | 112.9 |
| | | | T/C | 22 (34.4) | 16 (57.1) | 2.55 (1.03–6.32) | | |
| | | Log-additive | - | - | - | 1.06 (0.56–1.98) | 0.87 | 117 |
| CCL2 | rs1024611 (A/G) | Codominant | A/A | 35 (54.7) | 15 (53.6) | 1.00 | 0.99 | 119.1 |
| | | | G/A | 27 (42.2) | 12 (42.9) | 1.04 (0.42–2.58) | | |
| | | | G/G | 2 (3.1) | 1 (3.6) | 1.17 (0.10–13.87) | | |
| | | Dominant | A/A | 35 (54.7) | 15 (53.6) | 1.00 | 0.92 | 117.1 |
| | | | G/A-G/G | 29 (45.3) | 13 (46.4) | 1.05 (0.43–2.55) | | |
| | | Recessive | A/A-G/A | 62 (96.9) | 27 (96.4) | 1.00 | 0.91 | 117.1 |
| | | | G/G | 2 (3.1) | 1 (3.6) | 1.15 (0.10–13.21) | | |
| | | Overdominant | A/A-G/G | 37 (57.8) | 16 (57.1) | 1.00 | 0.95 | 117.1 |
| | | | G/A | 27 (42.2) | 12 (42.9) | 1.03 (0.42–2.52) | | |
| | | Log-additive | - | - | - | 1.05 (0.48–2.31) | 0.9 | 117.1 |
| DC-SIGN | rs735240 (A/G) | Codominant | G/G | 24 (37.5) | 11 (39.3) | 1.00 | 0.99 | 119 |
| | | | G/A | 26 (40.6) | 11 (39.3) | 0.92 (0.34–2.52) | | |
| | | | A/A | 14 (21.9) | 6 (21.4) | 0.94 (0.28–3.08) | | |
| | | Dominant | G/G | 24 (37.5) | 11 (39.3) | 1.00 | 0.87 | 117 |
| | | | G/A-A/A | 40 (62.5) | 17 (60.7) | 0.93 (0.37–2.31) | | |
| | | Recessive | G/G-G/A | 50 (78.1) | 22 (78.6) | 1.00 | 0.96 | 117.1 |
| | | | A/A | 14 (21.9) | 6 (21.4) | 0.97 (0.33–2.87) | | |
| | | Overdominant | G/G-A/A | 38 (59.4) | 17 (60.7) | 1.00 | 0.9 | 117.1 |
| | | | G/A | 26 (40.6) | 11 (39.3) | 0.95 (0.38–2.34) | | |
| | | Log-additive | - | - | - | 0.96 (0.53–1.73) | 0.9 | 117.1 |
| TLR2 | rs5743708 (A/G) | - | G/G | 58 (90.6) | 24 (85.7) | 1.00 | 0.5 | 116.6 |
| | | | G/A | 6 (9.4) | 4 (14.3) | 1.61 (0.42–6.23) | | |

(Continued)

**Table 7.** (Continued)

| Gene | dbSNP ID number[a] | Genetic Model | Genotype | Without thrombocytopenia n = 64 | With thrombocytopenia n = 28 | OR (95% CI) | P-value[b] | AIC |
|------|-------------------|---------------|----------|--------------------------------|------------------------------|-------------|-----------|-----|
| TLR4 | rs4986791 (C/T) | - | C/C | 59 (92.2) | 24 (85.7) | 1.00 | 0.35 | 116.2 |
|      |                 |   | T/C | 5 (7.8) | 4 (14.3) | 1.97 (0.49–7.96) |      |      |
| TLR9 | rs352140 (C/T) | Codominant | T/T | 19 (29.7) | 11 (39.3) | 1.00 | 0.5 | 117.7 |
|      |                |            | T/C | 33 (51.6) | 14 (50) | 0.73 (0.28–1.93) |     |      |
|      |                |            | C/C | 12 (18.8) | 3 (10.7) | 0.43 (0.10–1.87) |     |      |
|      |                | Dominant | T/T | 19 (29.7) | 11 (39.3) | 1.00 | 0.37 | 116.3 |
|      |                |          | T/C-C/C | 45 (70.3) | 17 (60.7) | 0.65 (0.26–1.65) |     |      |
|      |                | Recessive | T/T-T/C | 52 (81.2) | 25 (89.3) | 1.00 | 0.32 | 116.1 |
|      |                |           | C/C | 12 (18.8) | 3 (10.7) | 0.52 (0.13–2.01) |     |      |
|      |                | Overdominant | T/T-C/C | 31 (48.4) | 14 (50.0) | 1.00 | 0.89 | 117 |
|      |                |              | T/C | 33 (51.6) | 14 (50.0) | 0.94 (0.39–2.28) |     |      |
|      |                | Log-additive | - | - | - | 0.68 (0.35–1.33) | 0.25 | 115.7 |

Data presented as number (%), OR, odds ratio; CI, confidence interval; NA, not applicable; AIC, Akaike information criterion; p-values below 0.05 are statistically significant; IL, Interleukin; CCL 2, C-C motif chemokine ligand 2; DC-SIGN, dendritic cell-specific ICAM-grabbing non-integrin; TLR, Toll-like receptor.

[a] SNP database (dbSNP) reference number (ID number).

[b] P-value for comparison between infants without thrombocytopenia and with thrombocytopenia in cCMV group.

[c] The best fitted model based on AIC.

IL12 increased the risk of autoimmune diseases [33, 34]. Certainly, further investigations are needed to confirm the relationship between IL12B rs3212227 polymorphism and the decreased risk of prematurity and evaluate the pathomechanism of its action.

Next, we observed the association between polymorphism of IL1B rs16944 and the reduced risk of splenomegaly. Previously, there were some attempts to find out if splenomegaly is associated with SNPs in other diseases, but none of examined polymorphism predicted splenomegaly development [35]. The spleen, the lymphoid organ and reservoir of myeloid cells, was proved to mobilize neutrophils and inflammatory monocytes/macrophages to the blood when affected by the presence of IL-1β in multiple sclerosis [36]. Additionally, the expression of IL1B mRNA was detected in the spleen [37]. Until now, IL-1β polymorphisms were linked to some other diseases, including pediatric *Helicobacter pylori* infection, septic shock and seasonal influenza A/H3N2 virus infection [38].

In our study, 88% patients with splenomegaly demonstrated severe thrombocytopenia. We found that infants carrying heterozygous T/C genotype of IL28B rs12979860 polymorphism demonstrated a significantly higher frequency of thrombocytopenia in comparison to C/C-T/T genotypes. It should be underlined that, this is the first report to connect SNP of IL28B rs12979860 and increased risk of thrombocytopenia in cCMV infection. On the other hand, Thomas et al. documented that C/C genotype of IL28B rs12979860 was a predictor of virus elimination in patients with HCV infection [39]. By contrast, protective effect of the T allele of IL28B rs12979860 polymorphism was documented against HCMV infection in the allogeneic stem cell transplant adult patients [8].

Finally, we found that TLR4 rs4986791 polymorphism was related to hepatitis in newborn infants with cCMV infection, but the pathomechanism of this finding is unclear. In previous reports, TLR4 rs4986791 polymorphism provoked the disruption of the normal structure of the extracellular domain of TLR4, what might generate a protein with reduced binding affinity to the ligand, what was reported as a protective factor against *Helicobacter pylori* infection among children with gastritis [40, 41]. Additionally, Bochud et al. showed that TLR4 rs4986791 (C/T T399I) had a protective effect against leprosy [42]. Nevertheless, ours is the

Table 8. Associations between SNPs and hepatitis in newborn infants with cCMV infection.

| Gene | dbSNP IDnumber[a] | Genetic Model | Genotype | Without hepatitis n = 84 | With hepatitis n = 8 | OR (95% CI) | P-value[b] | AIC |
|---|---|---|---|---|---|---|---|---|
| IL1B | rs16944 (G/A) | Codominant | G/G | 32 (38.1) | 4 (50.0) | 1.00 | 0.46 | 58.8 |
| | | | A/G | 45 (53.6) | 4 (50.0) | 0.71 (0.17–3.06) | | |
| | | | A/A | 7 (8.3) | 0 (0.0) | 0.00 (0.00-NA) | | |
| | | Dominant | G/G | 32 (38.1) | 4 (50.0) | 1.00 | 0.51 | 57.9 |
| | | | A/G-A/A | 52 (61.9) | 4 (50.0) | 0.62 (0.14–2.63) | | |
| | | Recessive | G/G-A/G | 77 (91.7) | 8 (100.0) | 1.00 | 0.25 | 57 |
| | | | A/A | 7 (8.3) | 0 (0.0) | 0.00 (0.00-NA) | | |
| | | Overdominant | G/G-A/A | 39 (46.4) | 4 (50.0) | 1.00 | 0.85 | 58.3 |
| | | | A/G | 45 (53.6) | 4 (50.0) | 0.87 (0.20–3.70) | | |
| | | Log-additive | - | - | - | 0.56 (0.16–2.00) | 0.36 | 57.5 |
| IL12B | rs3212227 (G/T) | Codominant | T/T | 51 (60.7) | 6 (75.0) | 1.00 | 0.45 | 58.8 |
| | | | T/G | 26 (30.9) | 2 (25.0) | 0.65 (0.12–3.47) | | |
| | | | G/G | 7 (8.3) | 0 (0.0) | 0.00 (0.00-NA) | | |
| | | Dominant | T/T | 51 (60.7) | 6 (75.0) | 1.00 | 0.41 | 57.7 |
| | | | T/G-G/G | 33 (39.3) | 2 (25.0) | 0.52 (0.10–2.71) | | |
| | | Recessive | T/T-T/G | 77 (91.7) | 8 (100.0) | 1.00 | 0.25 | 57 |
| | | | G/G | 7 (8.3) | 0 (0.0) | 0.00 (0.00-NA) | | |
| | | Overdominant | T/T-G/G | 58 (69) | 6 (75.0) | 1.00 | 0.72 | 58.2 |
| | | | T/G | 26 (30.9) | 2 (25.0) | 0.74 (0.14–3.93) | | |
| | | Log-additive | - | - | - | 0.49 (0.11–2.14) | 0.3 | 57.3 |
| IL28B | rs12979860 (C/T) | Codominant | C/C | 40 (47.6) | 1 (12.5) | 1.00 | 0.093 | 55.6 |
| | | | T/C | 32 (38.1) | 6 (75) | 7.50 (0.86–65.52) | | |
| | | | T/T | 12 (14.3) | 1 (12.5) | 3.33 (0.19–57.39) | | |
| | | Dominant | C/C | 40 (47.6) | 1 (12.5) | 1.00 | 0.041 | 54.2 |
| | | | T/C-T/T | 44 (52.4) | 7 (87.5) | 6.36 (0.75–54.01) | | |
| | | Recessive | C/C-T/C | 72 (85.7) | 7 (87.5) | 1.00 | 0.89 | 58.3 |
| | | | T/T | 12 (14.3) | 1 (12.5) | 0.86 (0.10–7.60) | | |
| | | Overdominant | C/C-T/T | 52 (61.9) | 2 (25) | 1.00 | 0.043 | 54.3 |
| | | | T/C | 32 (38.1) | 6 (75) | 4.87 (0.93–25.63) | | |
| | | Log-additive | - | - | - | 1.89 (0.70–5.13) | 0.21 | 56.8 |
| CCL2 | rs1024611 (A/G) | Codominant | A/A | 48 (57.1) | 2 (25) | 1.00 | 0.14 | 56.5 |
| | | | G/A | 34 (40.5) | 5 (62.5) | 3.53 (0.65–19.28) | | |
| | | | G/G | 2 (2.4) | 1 (12.5) | 12.00 (0.74–194.64) | | |
| | | Dominant | A/A | 48 (57.1) | 2 (25) | 1.00 | 0.078 | 55.2 |
| | | | G/A-G/G | 36 (42.9) | 6 (75) | 4.00 (0.76–20.99) | | |
| | | Recessive | A/A-G/A | 82 (97.6) | 7 (87.5) | 1.00 | 0.22 | 56.9 |
| | | | G/G | 2 (2.4) | 1 (12.5) | 5.86 (0.47–72.91) | | |
| | | Overdominant | A/A-G/G | 50 (59.5) | 3 (37.5) | 1.00 | 0.23 | 56.9 |
| | | | G/A | 34 (40.5) | 5 (62.5) | 2.45 (0.55–10.94) | | |
| | | Log-additive | - | - | - | 3.49 (0.98–12.44) | 0.049 | 54.5 |
| DC-SIGN | rs735240 (A/G) | Codominant | G/G | 33 (39.3) | 2 (25.0) | 1.00 | 0.71 | 59.7 |
| | | | G/A | 33 (39.3) | 4 (50.0) | 2.00 (0.34–11.68) | | |
| | | | A/A | 18 (21.4) | 2 (25.0) | 1.83 (0.24–14.13) | | |
| | | Dominant | G/G | 33 (39.3) | 2 (25.0) | 1.00 | 0.41 | 57.7 |
| | | | G/A-A/A | 51 (60.7) | 6 (75.0) | 1.94 (0.37–10.20) | | |
| | | Recessive | G/G-G/A | 66 (78.6) | 6 (75.0) | 1.00 | 0.82 | 58.3 |
| | | | A/A | 18 (21.4) | 2 (25.0) | 1.22 (0.23–6.58) | | |
| | | Overdominant | G/G-A/A | 51 (60.7) | 4 (50.0) | 1.00 | 0.56 | 58 |
| | | | G/A | 33 (39.3) | 4 (50.0) | 1.55 (0.36–6.61) | | |
| | | Log-additive | - | - | - | 1.36 (0.53–3.51) | 0.53 | 58 |
| TLR2 | rs5743708 (A/G) | - | G/G | 75 (89.3) | 7 (87.5) | 1.00 | 0.88 | 58.3 |
| | | | G/A | 9 (10.7) | 1 (12.5) | 1.19 (0.13–10.81) | | |

(Continued)

**Table 8.** (Continued)

| Gene | dbSNP IDnumber[a] | Genetic Model | Genotype | Without hepatitis n = 84 | With hepatitis n = 8 | OR (95% CI) | P-value[b] | AIC |
|---|---|---|---|---|---|---|---|---|
| TLR4 | rs4986791 (C/T) | - | C/C | 78 (92.9) | 5 (62.5) | 1.00 | 0.024[c] | 53.2 |
| | | | T/C | 6 (7.1) | 3 (37.5) | 7.80 (1.49–40.81) | | |
| TLR9 | rs352140 (C/T) | Codominant | T/T | 27 (32.1) | 3 (37.5) | 1.00 | 0.22 | 57.4 |
| | | | T/C | 42 (50.0) | 5 (62.5) | 1.07 (0.24–4.85) | | |
| | | | C/C | 15 (17.9) | 0 (0.0) | 0.00 (0.00-NA) | | |
| | | Dominant | T/T | 27 (32.1) | 3 (37.5) | 1.00 | 0.76 | 58.3 |
| | | | T/C-C/C | 57 (67.9) | 5 (62.5) | 0.79 (0.18–3.55) | | |
| | | Recessive | T/T-T/C | 69 (82.1) | 8 (100.0) | 1.00 | 0.084 | 55.4 |
| | | | C/C | 15 (17.9) | 0 (0.0) | 0.00 (0.00-NA) | | |
| | | Overdominant | T/T-C/C | 42 (50.0) | 3 (37.5) | 1.00 | 0.5 | 57.9 |
| | | | T/C | 42 (50%) | 5 (62.5) | 1.67 (0.37–7.42) | | |
| | | Log-additive | - | - | - | 0.59 (0.19–1.83) | 0.35 | 57.5 |

Data presented as number (%), OR, odds ratio; CI, confidence interval; NA, not applicable; AIC, Akaike information criterion; p-values below 0.05 are statistically significant; IL, Interleukin; CCL 2, C-C motif chemokine ligand 2; DC-SIGN, dendritic cell-specific ICAM-grabbing non-integrin; TLR, Toll-like receptor.

[a] SNP database (dbSNP) reference number (ID number).

[b] P-value for comparison between infants without hepatitis and with hepatitis in cCMV group.

[c] The best fitted model based on AIC.

first report that links SNPs of TLR4 and hepatitis in cCMV infection. Additional studies are needed to confirm the relation of TLR4 polymorphism with increased risk of hepatitis among cCMV-infected infants.

The strength of our study is the prospective character of the research. Another advantage is the homogenous study population, containing only the Caucasian newborn infants. Furthermore, we limited our study group to newborns with confirmed cCMV infection only, in order to eliminate other potential genetic factors of postnatal HCMV infection. To the best of our knowledge, five out of eight SNPs genotyped in the current study have never been analyzed in the context of cCMV infection.

The present study has also some limitations. We are aware that hypothesis-driven candidate gene approach is essentially gene association study, however, it is also a subject to the artefacts of spurious association findings. Genes examined in the current study had been previously implied and some of them had been replicated poorly. However, most of these genes were not performed in newborn infants or if they were performed in that age group, the study populations were comparable or even smaller than ours.

What is more, it seems that different genes may take part in predisposition to the disease, while others host genes might influence on the clinical manifestation of the disease. We admit that the situation might be more complicated than it was previously assumed. Moreover, there are plenty of other external factors that could impact on the results: different HCMV strains, primary or secondary maternal infection, time of the fetus infection (which trimester). However, the current study was not designed to address these issues.

We acknowledge that the power of association study is improved if the sampling strategy takes account of exposure heterogenity, though this is not necessarily easy to do, especially in the setting of cCMV infection. In addition, no correction for multiple testing was applied, and so type I error cannot be completely ruled out.

Nevertheless, our study group of 92 cCMV-infected newborns is larger than in our previous research, we are aware that the study sample size might be still not large enough as for genetic studies. Undoubtedly, future studies on cCMV infection are demanded to replicate above-

mentioned findings in a much larger cohort. Moreover, there is a need for other studies of genetic characteristics of the unique and immature fetus-host and its predisposition to the infection, not only limiting to HCMV itself. We underline the necessity for meta-analytic studies in this subject.

To summarize, our study demonstrated new associations between SNPs in IL1B, IL12B, IL28B, TLR4 genes and symptoms of cCMV infection such as prematurity, splenomegaly, thrombocytopenia and hepatitis respectively. Further studies on the role of SNPs in the pathogenesis of cCMV infection and incorporation assessment of selected SNPs in the clinical practice are warranted.

## Supporting information

**S1 Table. Association between examined SNPs and abnormal MRI.** Data presented as number (%), OR, odds ratio; CI, confidence interval; NA, not applicable; NS, not significant (p-values above 0.05); MRI, magnetic resonance imaging; IL, Interleukin; CCL 2, C-C motif chemokine ligand 2; DC-SIGN, dendritic cell-specific ICAM-grabbing non-integrin; TLR, Toll-like receptor. [a] SNP database (dbSNP) reference number (ID number). [b] P-value for comparison between infants with normal MRI and abnormal MRI in cCMV group.
(DOCX)

**S2 Table. Association between examined SNPs and abnormal hearing.** Data presented as number (%), OR, odds ratio; CI, confidence interval; NA, not applicable; NS, not significant (p-values above 0.05); IL, Interleukin; CCL 2, C-C motif chemokine ligand 2; DC-SIGN, dendritic cell-specific ICAM-grabbing non-integrin; TLR, Toll-like receptor. [a] SNP database (dbSNP) reference number (ID number). [b] P-value for comparison between infants with normal hearing and abnormal hearing in cCMV group. [c] Abnormal hearing–was defined as air conduction thresholds > 20dBHL on the Auditory Brainstem Response (ABR) at least in one ear.
(DOCX)

**S3 Table. Association between examined SNPs and cholestasis.** Data presented as number (%), OR, odds ratio; CI, confidence interval; NA, not applicable; NS, not significant (p-values above 0.05); IL, Interleukin; CCL2, C-C motif chemokine ligand 2; DC-SIGN, dendritic cell-specific ICAM-grabbing non-integrin; TLR, Toll-like receptor. [a] SNP database (dbSNP) reference number (ID number). [b] P-value for comparison between infants without cholestasis and with cholestasis in cCMV group.
(DOCX)

**S4 Table. Association between examined SNPs and IUGR.** Data presented as number (%), OR, odds ratio; CI, confidence interval; NA, not applicable; NS, not significant (p-values above 0.05); IUGR, intrauterine growth restriction; IL, Interleukin; CCL 2, C-C motif chemokine ligand 2; DC-SIGN, dendritic cell-specific ICAM-grabbing non-integrin; TLR, Toll-like receptor. [a] SNP database (dbSNP) reference number (ID number). [b] P-value for comparison between infants without IUGR and with IUGR in cCMV group.
(DOCX)

**S5 Table. Association between examined SNPs and microcephaly.** Data presented as number (%), OR, odds ratio; CI, confidence interval; NA, not applicable; NS, not significant (p-values above 0.05); IL, Interleukin; CCL 2, C-C motif chemokine ligand 2; DC-SIGN, dendritic cell-specific ICAM-grabbing non-integrin; TLR, Toll-like receptor. [a] SNP database (dbSNP) reference number (ID number). [b] P-value for comparison between infants without

microcephaly and with microcephaly in cCMV group.
(DOCX)

**S6 Table. Association between examined SNPs and neutropenia.** Data presented as number (%), OR, odds ratio; CI, confidence interval; NA, not applicable; NS, not significant (p-values above 0.05); IL, Interleukin; CCL 2, C-C motif chemokine ligand 2; DC-SIGN, dendritic cell-specific ICAM-grabbing non-integrin; TLR, Toll-like receptor. [a] SNP database (dbSNP) reference number (ID number). [b] P-value for comparison between infants without neutropenia and with neutropenia in cCMV group.
(DOCX)

**S7 Table. Association between examined SNPs and chorioretinitis.** Data presented as number (%), OR, odds ratio; CI, confidence interval; NA, not applicable; NS, not significant (p-values above 0.05); IL, Interleukin; CCL 2, C-C motif chemokine ligand 2; DC-SIGN, dendritic cell-specific ICAM-grabbing non-integrin; TLR, Toll-like receptor. [a] SNP database (dbSNP) reference number (ID number). [b] P-value for comparison between infants without chorioretinitis and with chorioretinitis in cCMV group. [c] With chorioretinitis–if occurred at least in one eye.
(DOCX)

**S8 Table. Association between examined SNPs and petechiae.** Data presented as number (%), OR, odds ratio; CI, confidence interval; NA, not applicable; NS, not significant (p-values above 0.05); IL, Interleukin; CCL 2, C-C motif chemokine ligand 2; DC-SIGN, dendritic cell-specific ICAM-grabbing non-integrin; TLR, Toll-like receptor. [a] SNP database (dbSNP) reference number (ID number). [b] P-value for comparison between infants without petechiae and with petechiae in cCMV group.
(DOCX)

**S9 Table. Association between examined SNPs and hepatomegaly.** Data presented as number (%), OR, odds ratio; CI, confidence interval; NA, not applicable; NS, not significant (p-values above 0.05); IL, Interleukin; CCL 2, C-C motif chemokine ligand 2; DC-SIGN, dendritic cell-specific ICAM-grabbing non-integrin; TLR, Toll-like receptor. [a] SNP database (dbSNP) reference number (ID number). [b] P-value for comparison between infants without hepatomegaly and with hepatomegaly in cCMV group.
(DOCX)

## Acknowledgments

We are thankful to the clinical staff of Neonatal Intensive Care Unit for providing assistance in participants' assessment.

## Author Contributions

**Conceptualization:** Dominika Jedlińska-Pijanowska, Beata Kasztelewicz, Justyna Czech-Kowalska, Anna Dobrzańska.

**Data curation:** Dominika Jedlińska-Pijanowska, Maciej Jaworski, Klaudia Charusta-Sienkiewicz.

**Formal analysis:** Dominika Jedlińska-Pijanowska, Beata Kasztelewicz, Justyna Czech-Kowalska, Maciej Jaworski.

**Funding acquisition:** Anna Dobrzańska.

**Investigation:** Justyna Czech-Kowalska.

**Methodology:** Beata Kasztelewicz, Justyna Czech-Kowalska, Maciej Jaworski.

**Project administration:** Justyna Czech-Kowalska.

**Supervision:** Justyna Czech-Kowalska, Anna Dobrzańska.

**Writing – original draft:** Dominika Jedlińska-Pijanowska, Justyna Czech-Kowalska.

**Writing – review & editing:** Dominika Jedlińska-Pijanowska, Beata Kasztelewicz, Justyna Czech-Kowalska, Maciej Jaworski, Klaudia Charusta-Sienkiewicz, Anna Dobrzańska.

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
