## [Decision Letter · Decision Letter 0]

2 Mar 2020

PONE-D-20-04592

Association between single nucleotide polymorphisms (SNPs) of IL1, IL12, IL28 and TLR4 and symptoms of congenital cytomegalovirus infection.

PLOS ONE

Dear Prof. Czech-Kowalska,

Thank you for submitting your manuscript to PLOS ONE. After careful consideration, we feel that it has merit but does not fully meet PLOS ONE’s publication criteria as it currently stands. Therefore, we invite you to submit a revised version of the manuscript that addresses the points raised by two experts who reviewed your study.

Please address all points raised by the reviewers including the major concerns of Reviewer 1 who believes that the study approach is outdated and the manuscript should not be published. We will only be able to publish your work if these concerns are dispelled.

We would appreciate receiving your revised manuscript by Apr 16 2020 11:59PM. To enhance the reproducibility of your results, we recommend that if applicable you deposit your laboratory protocols in protocols.io, where a protocol can be assigned its own identifier (DOI) such that it can be cited independently in the future. For instructions see: http://journals.plos.org/plosone/s/submission-guidelines#loc-laboratory-protocols

We look forward to receiving your revised manuscript.

Kind regards,

Michael Nevels

Academic Editor

PLOS ONE

Journal Requirements:

2. We noticed that you have chosen the subsection category “[FOR JOURNAL STAFF USE ONLY]” for your manuscript. Unfortunately, this is not a valid category. At this time, please choose one or more subsections that best represent the topic(s) of your study.

Reviewers' comments:

Reviewer's Responses to Questions

**Comments to the Author**

1. Is the manuscript technically sound, and do the data support the conclusions?

Reviewer #1: No

Reviewer #2: Yes

2. Has the statistical analysis been performed appropriately and rigorously? 

Reviewer #1: No

Reviewer #2: Yes

3. Have the authors made all data underlying the findings in their manuscript fully available?

Reviewer #1: Yes

Reviewer #2: No

4. Is the manuscript presented in an intelligible fashion and written in standard English?

Reviewer #1: Yes

Reviewer #2: Yes

5. Review Comments to the Author

Reviewer #1: The manuscript topic is an outdated approach, where the variants in the known immune genes are associated with the CMV infection. These genes were previously implied in numerous studies, and replicated poorly, suggesting lack of real significance or at the least more complicated situation that previously assumed. Also, the focus of the clinically manifest disease will likely depend on the host genes, causing more selection bias in the study design. Lack of significance and traces of related phenotypes support this, and suggest an accidental finding (due to lack of multiple-testing adjustment, which is mandatory due to various genetic models being used in the analysis). The main hypothesis of this study should not be so focused on CMV itself (since the vast majority of people are positive), but to the underlying genetic characteristics that predispose some of these infected people/neonates towards infection. This is likely to be achieved through meta-analytic study. The current study is substantially underpowered, and hence of very limited contribution.

Reviewer #2: Authors should address the following points:

1. Table 1: summation of cCMV group (n=49) and control group (n=70) should be 119 not 129 for Cesarean section

2. In all tables the foot notes should clearly state to which groups’ statistical comparisons the P values belong to. (i.e. cCMV and control groups)

3. For tables where SNP genotypes are presented, for each SNP the change should be given. For example rs16944 (G/A)

4. Either in the Materials or in Introduction section why these candidate genes was chosen should be briefly explained (i.e. cytokines X,Y,Z were chosen for their pro-inflammatory role, etc.. ). This will also make the manuscript look more hypothesis driven.

5. AIC values presented in table 5 doesn’t convey much information as the reader doesn’t know the AIC values for other alternate models / effects of other not presented SNPs on a specific symptom.

6. Data on HCMV viral load, and maternal genotypes would also be interesting to examine

7. Thorough language, word usage, spelling check is advised.

6. PLOS authors have the option to publish the peer review history of their article (what does this mean?). If published, this will include your full peer review and any attached files.

Reviewer #1: No

Reviewer #2: No

---

## [Author Response · Author response to Decision Letter 0]

15 Apr 2020

We are pleased to resubmit the revised version of our manuscript PONE-D-20-04592 entitled „Association between single nucleotide polymorphisms (SNPs) of IL1, IL12, IL28 and TLR4 and symptoms of congenital cytomegalovirus infection”.

We would like to thank the reviewers for their constructive comments. We have introduced changes according to the reviewers’ comments. All changes are highlighted in yellow in marked-up version of the article.

Please find our response to the reviewers’ comments, below: 

Reviewer #1: The manuscript topic is an outdated approach, where the variants in the known immune genes are associated with the CMV infection. These genes were previously implied in numerous studies, and replicated poorly, suggesting lack of real significance or at the least more complicated situation that previously assumed. Also, the focus of the clinically manifest disease will likely depend on the host genes, causing more selection bias in the study design. Lack of significance and traces of related phenotypes support this, and suggest an accidental finding (due to lack of multiple-testing adjustment, which is mandatory due to various genetic models being used in the analysis). The main hypothesis of this study should not be so focused on CMV itself (since the vast majority of people are positive), but to the underlying genetic characteristics that predispose some of these infected people/neonates towards infection. This is likely to be achieved through meta-analytic study. The current study is substantially underpowered, and hence of very limited contribution.

Reply: We thank the reviewer for this important comment. Our study was single-centre and prospective one, conducted between 2016 and 2019 in homogenous population of Caucasian newborns. The methodology has been chosen based on current trends in the field of SNP associations with infectious diseases. Both hypothesis-driven candidate gene approach and genome-wide associations are essentially gene association studies as well as both are subject to the same artefacts of spurious association findings. In our study the former approach was chosen based on our previous expertise in the field and available resources. We used customized TaqMan Assay (rather than RLFP-PCR) which provide accurate and reproductive results. We acknowledged that the power of association study is improved if the sampling strategy takes account of exposure heterogeneity, though this is not necessarily easy to do, especially in the setting of congenital CMV infection. We also acknowledge the lack of correction for multi testing and these limitations were mentioned in the manuscript. 

We appreciate the reviewer’s point of view that the main hypothesis of this study should not be so focused on CMV itself (since the vast majority of people are positive). We agree that the vast majority of people are positive. The prevalence of CMV seropositive women in reproductive age is about 80% in Poland (Siennicka_ High Seroprevalence of CMV Among Women of Childbearing Age Implicates High Burden of Congenital Cytomegalovirus Infection in Poland_ Polisch J Microbiol 2016). The prevalence of seropositive newborns would be similar - most of newborn infants are seropositive due to transplacental transfer of maternal IgG but not to congenital infection. Nevertheless, a rate of cCMV is only about 0,5 % (in Poland estimated rate is 2,2 to 3,7%). Additionally, immaturity of fetal immune system makes this population very unique and may influence on the role of immune genes variations in congenital infection. Therefore we decided to examine the association between SNPs in genes encoding selected cytokines and cytokine receptors in newborns (also preterms) with proven congenital CMV infection. We agree that these genes were previously implied in numerous studies, and replicated poorly, however most of them were not performed in newborn infants, and if they were performed, the study populations were comparable or even smaller than ours. We underlined the need for further studies in larger population in our manuscript. We also agree that meta-analysis incorporating published SNP genotyping data is warranted to evaluate factors which predispose neonates towards CMV infection, thus data collected in studies like ours is desirable.

Reviewer #2:

1. Table 1: summation of cCMV group (n=49) and control group (n=70) should be 119 not 129 for Cesarean section

Reply: We would like to thank the reviewer for attention to details. We have addressed this comment by correcting data in Table 1.

2. In all tables the foot notes should clearly state to which groups’ statistical comparisons the P values belong to. (i.e. cCMV and control groups)

Reply: We have followed reviewer’s recommendation and we are very grateful for this suggestion. We have added suitable information in the foot notes to state which groups’ statistical comparison were presented. We have presented that information by using “b” superscript in all footnotes.

3. For tables where SNP genotypes are presented, for each SNP the change should be given. For example rs16944 (G/A)

Reply: We would like to thank the reviewer for attention to details. We have supplemented the tables with information about the change in nucleotides for each SNP gene.

4. Either in the Materials or in Introduction section why these candidate genes was chosen should be briefly explained (i.e. cytokines X,Y,Z were chosen for their pro-inflammatory role, etc.. ). This will also make the manuscript look more hypothesis driven.

Reply: In the Material and Methods section, we have added additional section called “Candidate genes selection”, where we provided rationale for selecting individual SNPs for the study. We have supported it with additional appropriate references.

5. AIC values presented in table 5 doesn’t convey much information as the reader doesn’t know the AIC values for other alternate models / effects of other not presented SNPs on a specific symptom.

Reply: We appreciate the reviewer’s precise analysis of the presented data. We understand that our presentation of data within Table 5 was not adequately detailed. Following the reviewer’s comments, we have made all the data underlying our findings fully available for the reader. Thus, we have moved four tables concerning symptoms with statistical significant findings from the Supplementary File to the main article. We have created four separate tables (Table 5-8) for each symptom instead of combined Table 5 from the original version of the article. 

In all new Tables, AIC values have been added, in order to allow readers to see the results of other alternate models. The best fitted model based on AIC for each SNP have been presented by using “ c” superscript in footnotes in all four new Tables (5 to 8). 

As the supporting information, we left nine tables (Tables S1-9) concerning data associations between examined SNPs and other analyzed symptoms among cCMV group, where no statistical finding were observed. 

We hope, that new presentation of our results is now more informative for the reader.

6. Data on HCMV viral load, and maternal genotypes would also be interesting to examine.

As far as HCMV viral load is concerned, we have put these data into separate article which is now being reviewed in other medical journal. 

We agree that maternal genotypes would be interesting to explore but our current project was not designed to address those issues. This was dictated by the fact that the present study was a single-center and conducted in hospital with no maternity unit. However, we strongly agree that it would be worth to incorporate maternal genotypes into a new multicenter project in the future.

7. Thorough language, word usage, spelling check is advised.

We have proofread the paper to eliminate all languages spelling errors.

---

## [Decision Letter · Decision Letter 1]

21 Apr 2020

PONE-D-20-04592R1

Association between single nucleotide polymorphisms (SNPs) of IL1, IL12, IL28 and TLR4 and symptoms of congenital cytomegalovirus infection.

PLOS ONE

Dear Prof. Czech-Kowalska,

Thank you for submitting your revised manuscript to PLOS ONE. We appreciate that the manuscript has been improved substantially. Please address the remaining minor points made by Reviewer 1 before we proceed to publication.

We would appreciate receiving your revised manuscript by Jun 05 2020 11:59PM. To enhance the reproducibility of your results, we recommend that if applicable you deposit your laboratory protocols in protocols.io, where a protocol can be assigned its own identifier (DOI) such that it can be cited independently in the future. For instructions see: http://journals.plos.org/plosone/s/submission-guidelines#loc-laboratory-protocols

We look forward to receiving your revised manuscript.

Kind regards,

Michael Nevels

Academic Editor

PLOS ONE

Reviewers' comments:

Reviewer's Responses to Questions

**Comments to the Author**

1. If the authors have adequately addressed your comments raised in a previous round of review and you feel that this manuscript is now acceptable for publication, you may indicate that here to bypass the “Comments to the Author” section, enter your conflict of interest statement in the “Confidential to Editor” section, and submit your "Accept" recommendation.

Reviewer #1: All comments have been addressed

Reviewer #2: All comments have been addressed

2. Is the manuscript technically sound, and do the data support the conclusions?

Reviewer #1: Yes

Reviewer #2: Yes

3. Has the statistical analysis been performed appropriately and rigorously? 

Reviewer #1: No

Reviewer #2: Yes

4. Have the authors made all data underlying the findings in their manuscript fully available?

Reviewer #1: Yes

Reviewer #2: Yes

5. Is the manuscript presented in an intelligible fashion and written in standard English?

Reviewer #1: Yes

Reviewer #2: Yes

6. Review Comments to the Author

Reviewer #1: Thank you; you have not provided responses to all of my statements; please, at the very least address some of these issues in the limitations section

Reviewer #2: Authors addressed the comments and corrections from both reviewers. Minor typos and grammatical issues can be handled by the journal.

7. PLOS authors have the option to publish the peer review history of their article (what does this mean?). If published, this will include your full peer review and any attached files.

Reviewer #1: No

Reviewer #2: Yes: Efe Sezgin

---

## [Author Response · Author response to Decision Letter 1]

27 Apr 2020

We are pleased to resubmit the revised version of our manuscript PONE-D-20-04592R1 entitled „Association between single nucleotide polymorphisms (SNPs) of IL1, IL12, IL28 and TLR4 and symptoms of congenital cytomegalovirus infection”.

We would like to thank the reviewers for their constructive comments. We have tried to introduced changes according to the reviewers’ comments. Please find our response to the reviewers’ comments, below: 

Reviewer #1: Thank you; you have not provided responses to all of my statements; please, at the very least address some of these issues in the limitations section

Reply: We thank the Reviewer for this comment to expand the limitation section in our article. Now, we hope that constructive remarks in this section make the article much more valuable for the readers and researchers. The improved limitation section is as below:

“The present study has also some limitations. We are aware that hypothesis-driven candidate gene approach is essentially gene association study, however, it is also a subject to the artefacts of spurious association findings. Genes examined in the current study had been previously implied and some of them had been replicated poorly. However, most of these genes were not performed in newborn infants or if they were performed in that age group, the study populations were comparable or even smaller than ours. 

What is more, it seems that different genes may take part in predisposition to the disease, while others host genes might influence on the clinical manifestation of the disease. We admit that the situation might be more complicated than it was previously assumed. Moreover, there are plenty of other external factors that could impact on the results: different HCMV strains, primary or secondary maternal infection, time of the fetus infection (which trimester). However, the current study was not designed to address these issues. 

We acknowledge that the power of association study is improved if the sampling strategy takes account of exposure heterogenity, though this is not necessarily easy to do, especially in the setting of cCMV infection. In addition, no correction for multiple testing was applied, and so type I error cannot be completely ruled out. 

Nevertheless, our study group of 92 cCMV-infected newborns is larger than in our previous research, we are aware that the study sample size might be still not large enough as for genetic studies. Undoubtedly, future studies on cCMV are demanded to replicate above-mentioned findings in a much larger cohort. Moreover, there is a need for other studies of genetic characteristics of the unique and immature fetus-host and its predisposition to the infection, not only limiting to HCMV itself. We underline the necessity for meta-analytic studies in this subject. “

---

## [Editor Report · Decision Letter 2]

29 Apr 2020

Association between single nucleotide polymorphisms (SNPs) of IL1, IL12, IL28 and TLR4 and symptoms of congenital cytomegalovirus infection.

PONE-D-20-04592R2

Dear Dr. Czech-Kowalska,

We are pleased to inform you that your manuscript has been judged scientifically suitable for publication and will be formally accepted for publication once it complies with all outstanding technical requirements.

With kind regards,

Michael Nevels

Academic Editor

PLOS ONE
---

## [Editor Report · Acceptance letter]

7 May 2020

PONE-D-20-04592R2 

Association between single nucleotide polymorphisms (SNPs) of IL1, IL12, IL28 and TLR4 and symptoms of congenital cytomegalovirus infection. 

Dear Dr. Czech-Kowalska:

I am pleased to inform you that your manuscript has been deemed suitable for publication in PLOS ONE. Congratulations! Your manuscript is now with our production department. 

With kind regards,

on behalf of

Dr. Michael Nevels 

Academic Editor

PLOS ONE